# Clinical Features and Outcome of Multidrug-Resistant Osteoarticular Tuberculosis: A 12-Year Case Series from France

**DOI:** 10.3390/microorganisms10061215

**Published:** 2022-06-14

**Authors:** Isabelle Bonnet, Elie Haddad, Lorenzo Guglielmetti, Pascale Bémer, Louis Bernard, Anne Bourgoin, Rachel Brault, Gaud Catho, Eric Caumes, Lélia Escaut, Eric Fourniols, Mathilde Fréchet-Jachym, Alice Gaudart, Hélène Guillot, Barthélémy Lafon-Desmurs, Jean-Philippe Lanoix, Philippe Lanotte, Adrien Lemaignen, Bénédicte Lemaire, Nadine Lemaitre, Christophe Michau, Philippe Morand, Faiza Mougari, Dhiba Marigot-Outtandy, Solène Patrat-Delon, Thomas Perpoint, Caroline Piau, Valérie Pourcher, Virginie Zarrouk, Valérie Zeller, Nicolas Veziris, Stéphane Jauréguiberry, Alexandra Aubry

**Affiliations:** 1Cimi-Paris, INSERM, U1135, Centre d’Immunologie et des Maladies Infectieuses, Sorbonne Université, 75013 Paris, France; isabelle.bonnet2@aphp.fr (I.B.); lorenzo.guglielmetti@aphp.fr (L.G.); nicolas.veziris@sorbonne-universite.fr (N.V.); 2Centre National de Référence des Mycobactéries et de la Résistance des Mycobactéries aux Antituberculeux, Hôpital Pitié-Salpêtrière, Assistance Publique-Hôpitaux de Paris (AP-HP), Sorbonne-Université, 75013 Paris, France; 3TB Consilium of the National Reference Center for Mycobacteria, 75013 Paris, France; 4Service de Maladies Infectieuses et Tropicales, Hôpital Pitié-Salpêtrière, AP-HP, INSERM 1136, Sorbonne-Université, 75013 Paris, France; ehaddad@live.com (E.H.); eric.caumes@aphp.fr (E.C.); valerie.martinez@aphp.fr (V.P.); stephane.jaureguiberry@aphp.fr (S.J.); 5Department of Bacteriology, University Hospital, CHU Nantes, 44000 Nantes, France; pascale.bemer@chu-nantes.fr; 6Service de Médecine Interne et Maladies Infectieuses, Centre Hospitalier Régional Universitaire de Tours, 37000 Tours, France; prlbernard@orange.fr (L.B.); adrien.lemaignen@chu-tours.fr (A.L.); 7Service de Virologie et Mycobactériologie, Centre Hospitalier Universitaire de Poitiers, 86000 Poitiers, France; anne.bourgoin@chu-poitiers.fr; 8Service de Rhumatologie, Centre Hospitalier Universitaire de Poitiers, 86000 Poitiers, France; rachel.brault@chu-poitiers.fr; 9Service de Maladies Infectieuses et Tropicales, Hospices Civils de Lyon, 69002 Lyon, France; gaud.catho@hcuge.ch (G.C.); thomas.perpoint@chu-lyon.fr (T.P.); 10Service de Maladies Infectieuses et Tropicales, Hôpital Bicêtre, AP-HP, Université Paris Saclay, 94270 Le Kremlin-Bicêtre, France; lelia.escaut@aphp.fr; 11Service de Chirurgie Orthopédique, Hôpital Pitié-Salpêtrière, AP-HP, Sorbonne-Université, 75013 Paris, France; eric.fourniols@aphp.fr; 12Sanatorium, Centre Hospitalier de Bligny, 91640 Briis-sous-Forges, France; m.jachym@chbligny.fr (M.F.-J.); b.lemaire@chbligny.fr (B.L.); d.marigotouttandy@chbligny.fr (D.M.-O.); 13Service de Bactériologie, Centre Hospitalier Universitaire de Nice, 06000 Nice, France; gaudart.a@chu-nice.fr; 14Service de Médecine Interne, Hôpital Robert Ballanger, 93600 Aulnay-sous-Bois, France; heleneguillot@hotmail.com; 15Service de Maladies Infectieuses et Tropicales, Centre Hospitalier de Tourcoing, 59200 Tourcoing, France; blafondesmurs@ch-tourcoing.fr; 16Service de Maladies Infectieuses et Tropicales, Centre Hospitalier Universitaire d’Amiens-Picardie, 80054 Amiens, France; lanoix.jean-philippe@chu-amiens.fr; 17Service de Bactériologie, Centre Hospitalier Universitaire de Tours, 37000 Tours, France; philippe.lanotte@univ-tours.fr; 18Service de Bactériologie, Centre Hospitalier Universitaire d’Amiens-Picardie, 59200 Tourcoing, France; lemaitre.nadine@chu-amiens.fr; 19Service de Maladies Infectieuses et Tropicales, Centre Hospitalier de Saint-Nazaire, 44606 Saint-Nazaire, France; c.michau@ch-saintnazaire.fr; 20Service de Bactériologie, Hôpital Cochin, AP-HP, Centre-Université de Paris, 75014 Paris, France; philippe.morand@aphp.fr; 21Service de Bactériologie, Hôpital Lariboisière, AP-HP, Nord-Université de Paris, 75018 Paris, France; faiza.mougari@aphp.fr; 22Service de Maladies Infectieuses et Tropicales, Centre Hospitalier Universitaire de Rennes, 35033 Rennes, France; solene.patrat-delon@chu-rennes.fr; 23Service de Bactériologie, Centre Hospitalier Universitaire de Rennes, 35033 Rennes, France; caroline.piau@chu-rennes.fr; 24Service de Médecine Interne, Hôpital Beaujon, AP-HP, Nord-Université de Paris, 92110 Clichy, France; virginie.zarrouk@aphp.fr; 25Centre de Référence des Infections Ostéo-Articulaires Complexes, Groupe Hospitalier Diaconesses Croix Saint-Simon, 75020 Paris, France; vzeller@hopital-dcss.org; 26Service de Bactériologie, Hôpitaux Saint-Antoine, Tenon, Trousseau, Rothschild, AP-HP, 75012 Paris, France; 27Centre de Référence des Infections Ostéo-Articulaires Complexes, Hôpital Pitié-Salpêtrière, AP-HP, 75013 Paris, France

**Keywords:** MDR-TB, XDR-TB, bone, spinal

## Abstract

The optimal treatment for osteoarticular infection due to multidrug-resistant tuberculosis strains (MDR-OATB) remains unclear. This study aims to evaluate the diagnosis, management and outcome of MDR-OATB in France. We present a case series of MDR-OATB patients reviewed at the French National Reference Center for Mycobacteria between 2007 and 2018. Medical history and clinical, microbiological, treatment and outcome data were collected. Twenty-three MDR-OATB cases were reported, representing 3% of all concurrent MDR-TB cases in France. Overall, 17 were male, and the median age was 32 years. Six patients were previously treated for TB, including four with first-line drugs. The most frequently affected site was the spine (*n* = 16). Bone and joint surgery were required in 12 patients. Twenty-one patients (91%) successfully completed the treatment with a regimen containing a mean of four drugs (range, 2–6) for a mean duration of 20 months (range, 13–27). Overall, high rates of treatment success were achieved following WHO MDR-TB treatment guidelines and individualized patient management recommendations by the French National TB Consilium. However, the optimal combination of drugs, duration of treatment and role of surgery in the management of MDR-OATB remains to be determined.

## 1. Introduction

Tuberculosis (TB) remains a major public health concern worldwide [1]. Extrapulmonary TB accounts for about one quarter of the TB cases, whereas osteoarticular TB (OATB) represents 10 to 15% of all extrapulmonary TB cases in Europe and the USA [2]. Sadly, there are no specific data for France. OATB onset is usually insidious and a long delay in diagnosis is common [2]. Spinal TB, which accounts for about 50% of OATB patients in most series, may lead to neurologic deficits and spinal deformity in up to 50% of cases [2,3]. In spinal TB, the thoracic and lumbar vertebrae are mostly involved, and multifocal non-contiguous bone destruction with relative disc preservation and paravertebral abscesses are common [3]. Extraspinal TB localizations are diverse, and prosthetic joint infections can also occur [4]. Overall, microbiologic confirmation of the diagnosis is rare, as samples are difficult to obtain [2].

According to the World Health Organization (WHO), in 2020, 150,000 people developed rifampin-resistant TB or multidrug-resistant TB (MDR-TB), defined as resistant to rifampin and isoniazid [1]. Due to disruptions in TB diagnosis and treatment during the COVID-19 outbreak, TB and MDR-TB cases are expected to increase globally [1]. Overall, MDR-TB-causing strains are estimated to be implicated in approximately 2% of all OATB cases [1,5].

Guidelines for the treatment of MDR-TB are based on those for pulmonary TB, as no specific recommendations for osteoarticular forms are available [6,7]. In this study, we present a 12-year national case series of MDR-OATB with the aim of describing the diagnosis, treatment, and outcome of these rare forms.

## 2. Materials and Methods

### 2.1. Study Population

We retrospectively included all consecutive bacteriologically-confirmed MDR-OATB cases reported to the French National Reference Center for Mycobacteria (NRC) from 1 January 2007 to 31 December 2018. The following data were retrieved from medical files, anonymized and collated in a database at the NRC: demographics, history of previous TB treatment, comorbidities, clinical presentation, TB localization, drug susceptibility testing results, treatment, adverse events, surgery and treatment outcome.

### 2.2. Definitions

Standard definitions of MDR- and XDR-TB were used (MDR-TB is a TB resistant to rifampin and isoniazid; XDR-TB is a MDR-TB also resistant to any fluoroquinolone and to at least one of three second-line injectable drugs (capreomycin, kanamycin and amikacin) [8]. The revised definition of XDR-TB released in 2021 was not used since the study includes cases of TB occurring before 2021 [9]. MDR-OATB cases refers to bacteriologically-confirmed cases of MDR-TB involving joints, bones or soft tissue adjacent to the affected bone or joint [8]. A bacteriologically-confirmed TB case is one from whom a biological specimen is positive by smear microscopy, culture or WHO-approved rapid diagnostics, i.e., genotypic methods.

Treatment outcome was assigned according to WHO definitions [8] and the proposal of Schwoebel et al. [10] in cases of treatment failure (Appendix A).

### 2.3. Drug Susceptibility Testing

Phenotypic drug susceptibility testing was performed using the proportion method [11]. Mutations involved in resistance to anti-TB drugs were identified by line probe assays (Genotype MTBDRplus and MTBDRsl, Hain Lifescience, Nehren, Germany) or in-house PCR combined with Sanger sequencing [12].

### 2.4. Statistical Analysis

Continuous variables are presented as mean (range) or median (interquartile range, IQR), and categorical variables as proportion. The rate of drug resistance was compared between MDR-OATB and overall MDR-TB strains using Fisher’s exact test. Statistical analysis was performed with Stata version 11.0 (StataCorp, College Station, TX, USA). Significance was determined as *p* < 0.05.

### 2.5. Ethical Considerations

The study protocol was approved by the Comité d’Ethique de la Recherche de Sorbonne Université (Approval Number: CER-2021-005) and registered on ClinicalTrials.gov (ID: APHP210051).

## 3. Results

### 3.1. Patient Characteristics 

Overall, 863 MDR-TB cases were reported to the NRC between 2007 and 2018; among these, 329 were extrapulmonary (alone or associated with pulmonary TB) and 23 were MDR-OATB cases (2.7% and 7.0% of all MDR-TB and of extrapulmonary MDR-TB cases, respectively) (Figure 1).

Among the 23 patients, the male/female sex ratio was 2.8/1 and the median age was 32 years (IQR, 25–40) (Table 1 and Table 2). Overall, six patients were previously treated for TB: four with first-line drugs, one with second-line drugs and one with an unknown treatment regimen. For the 23 patients, the median time from symptom onset to diagnosis was 3.5 months (IQR, 2–6). The most common clinical presentation was local pain (78.3%) and weight loss (52.2%). In most cases (*n* = 16), OATB affected the spine. In 14 patients, evidence of active TB was also found elsewhere, predominantly in the lung and lymph nodes (Table 1 and Table 2).

### 3.2. Methods Used for the Diagnosis of OATB 

All TB cases were bacteriologically-confirmed. OATB diagnosis was proven microbiologically for 19 patients and based on radiologic findings in four patients (Table 2). Positive cultures were obtained from bone or joint samples in 19 cases, and from other samples in four cases (Table 2). Histology was performed in less than half of the cases, even in cases of spinal TB, and was suggestive of TB (giant cell epithelioid granuloma with caseous necrosis) in eight out of nine patients (Table 2).

### 3.3. Drug Susceptibility Testing

Overall, resistance to antibiotics was significantly lower (*p* < 0.05) among MDR-OATB patients than among all MDR-TB patients diagnosed in France during the same period for streptomycin, ethambutol, pyrazinamide, kanamycin, cycloserine, fluoroquinolones, capreomycin and PAS (Table 3).

### 3.4. Medical and Surgical Treatment 

Seventeen patients were presented in the frame of the French TB Consilium [14]; the initial treatment regimen was based on genotypic resistance results, and then adapted according to phenotypic drug susceptibility testing results. For the last six patients, treatment was initiated in their setting and also based on genotypic and phenotypic resistance results. Patients received an average of four drugs in the intensive (range, 3–6) and the continuation phase of treatment (range, 2–5). All except two patients received two of the three drugs belonging to group A (fluoroquinolones (*n* = 21), linezolid (*n* = 21) and bedaquiline (*n* = 7)). Twenty-one patients completed the treatment with a mean duration of 20 months (range, 13–27 months), whereas two patients were not evaluated as they were lost to follow-up after 9 and 15 months of treatment, respectively. In 22 cases, a drug was withdrawn because of toxicity (Table 2).

Bone and joint surgery were performed for six patients with spinal TB and for six patients with other osteoarticular localizations. Surgical procedures were performed at the time of MDR-OATB diagnosis, except for the bone graft, the vertebral cementoplasty and one laminectomy that were performed 6 months after the initiation of treatment.

### 3.5. Patient Follow-Up and Treatment Outcome 

Overall, 21 out of 23 (91%) patients achieved treatment success. As previously proposed [10], we did not take into account the WHO criterion concerning discontinuation of ≥2 drugs in the case of adverse events. Interestingly, the five patients in whose case ≥2 drugs were discontinued because of adverse events achieved treatment success (Appendix A). Post-treatment follow-up was available for 14 patients, none of whom experienced failure or relapse. Sequelae, namely arthralgia, pain, sciatica and paresthesia, were reported in four patients during post-treatment follow-up. Fifteen patients performed imaging at the end of treatment: in 12 cases, persisting osteoarticular abnormalities were observed (Table 2).

## 4. Discussion

In our study, we report high rates of treatment success in MDR-OATB cases treated with individualized regimens in France. To date, our knowledge about the epidemiology and treatment outcome of MDR-OATB is limited, and mostly based on case reports and small retrospective series lacking comprehensive data on drug regimen and treatment outcome [15,16,17]. Moreover, WHO guidelines for the treatment of MDR-TB do not distinguish among different localizations of TB [6,7].

In our study, we have shown that MDR-OATB is rare among all MDR-TB cases (2.7%) and that overall high treatment success rates can be achieved by using individualized treatment following international recommendations and with the support of the French national TB Consilium [6,7,14]. Between 2007 and 2018, 2.7% and 7.0% of the MDR-TB patients diagnosed in France had an OATB among all MDR-TB cases and among the extrapulmonary cases, respectively. These proportions are similar to those previously reported [2,5]. Among our patients, the spine was affected in about 70% of all cases, while, at the time of MDR-OATB diagnosis, 61% of patients had active TB in at least one other localization, predominantly the lungs and lymph nodes. Similar findings have been reported in the literature [2,3,18].

Diagnosing OATB is challenging for several reasons: (i) the detection relies on specific tests, (ii) it is clinically indistinguishable from other chronic infections caused by more common bacterial pathogens, and (iii) the expertise in diagnosing TB has decreased in low-burden countries [18,19]. In the present study, the median time from symptom onset to OATB diagnosis was 3.5 months, which is longer than for overall TB cases in France (2 months) [19] but similar to what was reported recently for bone and joint TB in France [18].

The diagnosis of MDR-OATB was based mainly on microbiological findings. However, imaging was shown to be indispensable to establish the diagnosis of OATB in 17% of patients, highlighting the necessity of imaging for diagnosing extrapulmonary TB (Table 2). At the end of treatment, persisting osteoarticular abnormalities are usual; therefore, systematic imaging should be avoided. Indeed, using imaging to define treatment duration could lead clinicians to unnecessarily prolonged treatment with a higher risk of toxicity [20].

In our study, the rates of drug resistance in MDR-OATB strains were much lower than those of other MDR-TB strains in France (Table 3). An explanation might be that the majority of the 30% of patients with MDR-OATB with a previous active or latent TB were treated with the standard first-line TB regimen, whereas the majority of the 40% of the MDR-TB patients with a previous active TB were treated with second-line drugs (NRC data, not presented). As previously described, in the former cases, drug susceptibility profiles could be related to acquired drug resistance when patients were treated with first- or second-line drug regimens [7,21].

A daunting question concerning all osteoarticular infections is whether surgery is required, especially in cases of prosthetic infections. Although some evidence suggests that *M. tuberculosis* can develop biofilms [22], there are studies that document favorable outcomes of OATB without prosthesis removal [23,24,25]. MDR-TB guidelines are only available for lung surgery [6,7]. According to ATS/CDC/IDSA guidelines, surgery may be indicated in the case of failure to respond to chemotherapy with evidence of ongoing infection, relief of cord compression in patients with persistence or recurrence of neurologic deficits, or instability of the spine [20]. Such criteria could be reasonably applied to MDR-OATB cases. In our series, bone and joint surgery was performed for six patients with spinal TB, of whom four had neurological impairment, in accordance with the above indications [20], and for six patients with another osteoarticular localization, including one with a prosthesis and one with articular damage requiring a knee joint prosthesis.

Treatment regimens used in our study followed WHO MDR-TB treatment recommendations used before 2018, namely, the use of injectables as part of longer (18–20 months) individualized regimens [6,7]. Regimen design was supported by the national TB Consilium for 74% of patients [14]. Since 2016, WHO guidelines have recommended a shorter (9–11 months) regimen for patients with MDR-TB without resistance to drugs included in the regimen [6]; subsequently, the second-line injectable in the shorter regimen was replaced by bedaquiline, resulting in an all-oral regimen [26]. Retrospectively, five patients among our cohort would have been eligible for this shorter regimen. However, more studies are needed before applying the shorter regimen to MDR-OATB since (i) there is limited evidence on the activity of the shorter regimen in extrapulmonary TB in general, and in OATB in particular, and (ii) overall, while it is well-known that some drugs such as rifampin, fluoroquinolones and linezolid penetrate well into bone tissue [27], there is a lack of information regarding concentrations of new anti-TB drugs, such as bedaquiline.

Strong points of this study are the bacteriological confirmation of all cases and representative number of national MDR-TB patients included through the French National Reference Center [28]. This study also has some limitations. First, although one of the largest reported case series of MDR-OATB, the number of cases remains small. Second, post-treatment follow-up was not available for many patients, which makes it difficult to draw conclusions regarding patient management.

Overall, however, our study provides promising results in the treatment of MDR-OATB with longer, individualized regimens.

## Figures and Tables

**Figure 1 microorganisms-10-01215-f001:**
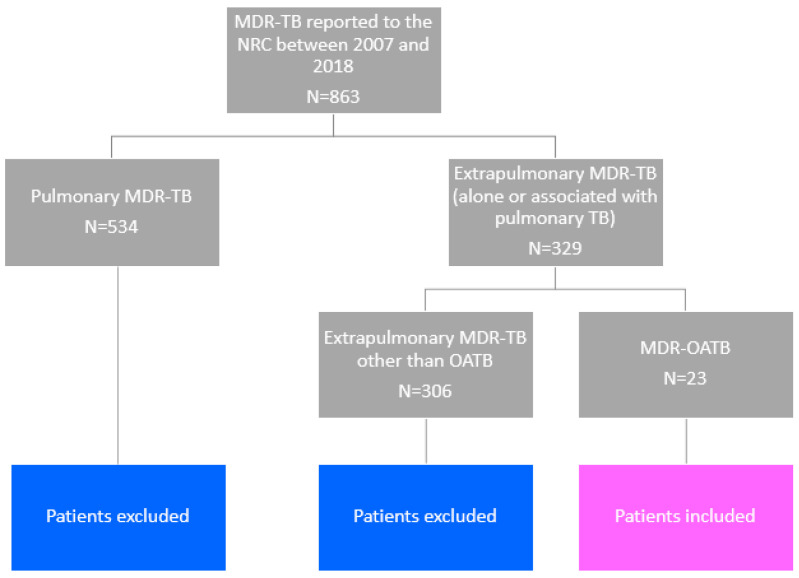
Flowchart for the study.

**Table 1 microorganisms-10-01215-t001:** Characteristics of 23 patients with multidrug-resistant osteoarticular TB in France from 2007 to 2018.

	*n*	%
**Sex**		
Female	6	26.1
Male	17	73.9
**Age (years)**		
0–14	1	4.3
15–24	4	17.4
25–44	16	69.7
45–64	1	4.3
≥65	1	4.3
**Country of birth**		
Western Europe	3	13.1
Africa	11	47.8
Eastern Europe	4	17.4
Asia	5	21.7
**Duration of stay in France for foreign-born patients (years)**		
<5	16	
5–10	4	
>10	1	
**Previous TB treatment** [8]		
New patients	17	73.9
Previously treated patients	6	26.1
Relapse	4	
Treatment after failure	1	
Other previously treated patients	1	
**Comorbidities ^1^**		
HIV infection	4	17.4
Hepatitis B infection	1	4.3
Hepatitis C infection	1	4.3
Chronic renal failure	1	4.3
Immunosuppressive therapy	1	4.3
**Body mass index < 18.5**	7	30.4
**Clinical signs/symptoms**		
Pain	18	78.3
Weight loss	12	52.2
Cough	6	26.1
Neurological deficit	5	21.7
Fever	5	21.7
**Median (IQR) C-reactive protein at diagnosis (mg/L)**	66 (21.4–114.3)	
**Median (IQR) serum vitamin D at diagnosis (ng/L)**	6 (5–14.5)	
**Delay between first symptoms and TB diagnosis (months)**		
<6	15	65.2
6–12	5	21.7
≥12	2	8.8
Unknown	1	4.3
**Tuberculin skin test**		
Positive (>10 mm)	7	30.4
Not done	16	69.6
**Interferon-gamma release assay**		
Positive	2	8.8
Undetermined	1	4.3
Not done	20	86.9
**Osteoarticular localization ^2^**		
Spine	16	69.6
Hip joint	4	17.4
Knee joint	2	8.8
Ribs	2	8.8
Sacro-iliac joint	3	13.1
Calcaneum	1	4.3
**Pulmonary TB associated**	9	39.1

^1^ Diabetes mellitus was not added to the table since no patient had diabetes. ^2^ Six patients had TB with multiple localizations.

**Table 2 microorganisms-10-01215-t002:** Clinical features of multidrug-resistant osteoarticular tuberculosis in France from 2007 to 2018.

Patient	Sex, Age (Years)	Comorbidities	Previous TB Treatment	Year of TB Diagnosis	Bone/Joint Localization	Other TB Localizations	Sample Used for Diagnosis	Histology of the Sample Used for Diagnosis	Treatment *	Surgical Treatment	Outcome (Post-Treatment Follow-Up) [8] ^§^
1	M, 42	None	None	2009	Left hip joint	-	Joint aspirate	ND	^1–2^[Am-Emb ^†^-Lzd ^†^-Mfx-PAS ^‡^]/ ^3^[Am-Mfx-PAS ^‡^]/ ^4–6^[Am ^†^-Cs -Mfx-PAS ^‡^]/ ^7–27^[Cs-Mfx-PAS ^‡^]	Debridement	Success ^§^ (5 years)
2	F, 65	HIV and chronic renal failure	Isoniazid-monoresistant TB 4 years previously	2009	Right knee joint	-	Joint aspirate	Inflammatory joint fluid	^1–16^[Emb-Mfx-Pza]	No surgery	Success (no post-treatment follow-up)
3	M, 41	None	None	2010	C5–C6, prevertebral abscess	Lymph nodes, liver, spleen, lung	Prevertebral abscess	ND	^1–2^[Am-Emb-Eto-Lzd-Mfx-Pza]/ ^3^[Am-Emb-Lzd-Mfx-Pza]/ ^4–5^[Emb-Lzd ^†^-Mfx-Pza]/ ^6–20^[Emb-Mfx-Pza]	Anterior vertebrectomy	Success (no post-treatment follow-up)
4	M, 48	None	None	2011	T2 until the conus of spinal cord, left iliac bone	Kidneys, meningitis, lung	Urine	ND	^1–3^[Am-Emb-Lzd ^†^-Mfx-Pza]/ ^4–6^[Am-Emb-Mfx-PAS ^†^-Pza]/ ^7–19^[Emb-Cs-Mfx-Pza]	No surgery	Success ^§^ (no post-treatment follow-up)
5	F, 21	None	None	2012	T3-T5	Lymph nodes	Lymph node	Epithelioid granuloma with caseous necrosis	^1^[Am-Lzd ^†^-Mfx]/ ^2–4^[Am-Bdq-Mfx ^†^-PAS]/ ^5^[Am-Lfx-PAS]/ ^6–8^[Am-Lfx-PAS ^†^]/ ^9–11^[Am-Lfx]/ ^12–19^[Cfz-Lfx]	No surgery	Success ^§^ (no post-treatment follow-up)
6	F, 40	Hepatitis C infection	Several episodes of TB the last 20 years	2013	Ribs at T5–T6 level	Lymph nodes, lung	Pulmonary sample	ND	^1^[Bdq-Cs-Emb-Lzd-Mpm/Clv-PAS]/ ^2–7^[Bdq ^†^-Cs-Emb-Lzd-Mpm/Clv]/ ^8–10^[Cs-Emb-Lzd-Mpm/Clv-Pza]/ ^11–25^[Cs-Emb-Lzd-Pza]	Lobectomy	Success (3 months)
7	M, 27	None	None	2013	C6, T8, L3, S1	Lung	Bone tissue	ND	^1^[Am-Cs-Lzd-Mpm/Clv-Mfx-PAS]/ ^2–6^[Am-Cs-Lzd-Mfx-PAS ^†^]/ ^7–25^[Cs-Lzd-Mfx-Pza]	Laminectomy	Success (2 years)
8	M, 30	None	Isoniazid-monoresistant TB 2 years previously	2013	T9–T12, prevertebral abscesses	-	Prevertebral abscess	ND	^1–4^[Am-Cs-Emb-Lzd-Mfx-Pza]/ ^5–10^[Cs-Emb-Lzd-Mfx-Pza ^†^]/ ^11–20^[Cs-Emb-Lzd-Mfx]	No surgery	Success (no post-treatment follow-up)
9	M, 29	None	None	2014	T12–L4, paravertebral abscess	-	Bone tissue	ND	^1–3^[Am-Lzd-Mfx-PAS]/ ^4–?^[Lzd-Mfx-PAS] (duration not known due to loss of follow-up)	Abscess drainage, stabilization	Not evaluated (loss of follow-up after 9 months of treatment)
10	F, 39	None	None	2014	Left hip prosthetic joint, psoas abscess	-	Synovial tissue	Epithelioid granuloma with giant cells and necrosis	^1–2^[Am-Cs-Lzd-Mfx-PAS]/ ^3–7^[Am-Lzd^‡^-Mfx-Pza-Rif]/ ^8–24^[Lzd^‡^-Mfx-Pza-Rif]	Revision arthroplasty (two-stage exchange)	Success (2 years)
11	M, 32	None	None	2014	L4–L5, L5–S1, right sacro-iliac joint	Lymph nodes, lung	Bone tissue	Granuloma and necrosis	^1–2^[Am-Lfx-Lzd-Pza]/ ^3–6^[Am-Lzd-Mfx-Pza-Rif]/ ^7–18^[Lzd-Mfx-Pza-Rif]	Vertebral cementoplasty	Success (6 years)
12	F, 39	None	Isoniazid-monoresistant TB 2 years before	2014	Right calcaneum	-	Bone tissue	ND	^1^[Am-Emb-Lzd-Mfx-Pza]/ ^2^[Am-Emb-Lzd-Mfx-Pza-Rif ^†^]/ ^3–6^[Am-Emb-Lzd-Mfx-Pza]/ ^7–12^[Emb ^†^Lzd ^†^-Mfx-Pza]/ ^13–19^[Cs-Mfx-PAS-Pza]	Resection then bone graft (after 1 year of antibiotic therapy)	Success ^§^ (3 years)
13	M, 15	None	None	2014	T10–L1, pre- and paravertebral abscesses	-	Paravertebral abscess	ND	^1–5^[Am-Bdq-Cs-Lfx-Lzd-Pza]/ ^6–7^[Bdq-Cs-Lfx-Lzd-Pza]/ ^8–9^[Cs-Lfx-Lzd-Pza]/ ^10–20^[Cs-Lfx-Lzd-Pza-Rif]/ ^21–24^[Lfx-Lzd-Pza]	No surgery	Success (no post-treatment follow-up)
14	M, 27	None	None	2015	T5, dorsal abscess	Lung	Paravertebral abscess	ND	^1–12^[Emb ^†^-Mfx-Pza-Rif]/ ^13–15^[Mfx-Pza-Rif]	No surgery	Success despite diffuse arthralgia (2 years)
15	M, 20	Rheumatoid arthritis (corticosteroids and methotrexate)	None	2015	L3–L4, paravertebral abscess	Lymph nodes	Bone tissue	Epithelioid granuloma with giant cells and necrosis	^1–6^[Am-Cs-Lzd-Mfx-PAS-Pza ^†^]/^7^[Am-Cs-Lzd-Mfx-PAS]/ ^8–19^[Cs-Lzd-Mfx-PAS]	No surgery	Success despite back pain (2 months)
16	F, 39	HIV	Drug-susceptible TB 3 years previously	2015	T11–T12, paravertebral abscess, left knee joint	-	Bone tissue	ND	^1–7^[Am-Emb-Lzd-Mfx-Pza]/ ^8–26^[Emb-Lzd-Mfx-Pza]	Resection arthroplasty	Success despite knee pain (3 years)
17	M, 31	None	None	2017	C6–C7, paravertebral abscess	Spleen	Bone tissue	Epithelioid granuloma with giant cells and caseous necrosis	^1–4^[Am-Cfx-Emb-Lzd-Mfx-Pza]/ ^5–19^[Cfx-Emb-Lzd-Mfx-Pza]	No surgery	Success despite left sciatica S1, cervical paresthesia (1 year)
18	M, 40	None	None	2018	C5, T10, T11, L1–L4, S1, pre- and paravertebral abscesses	Lung, lymph nodes, pleura, liver, spleen, small intestine	Bone tissue	Granuloma	^1^[Am-Emb-Eto-Lzd-Mfx-Pza]/ ^2–3^[Am ^†^-Bdq-Cfz-Lzd-Mfx-Pza]/ ^4–19^[Bdq-Cfz-Lzd-Mfx-Pza]	Corpectomy and anterior cervical arthrodesis	Success (6 months)
19	M, 25	None	None	2018	T4, T5, L1, L4, paravertebral and psoas abscesses	Lung, pleura	Bone tissue	Epithelioid granuloma with giant cells without necrosis	^1–2^[Am-Bdq-Cfz-Cs-Lzd-Mfx]/^3^[Bdq-Cfz-Cs-Lzd-Mfx]/ ^4–6^[Bdq-Cfz-Dlm-Lzd-Mfx]/ ^7–13^[Bdq-Cfz-Dlm-Mfx]	No surgery	Success (3 months)
20	M, 16	Hepatitis B infection	None	2018	L3, S3, right hip joint	Lymph nodes, pleura, liver, peritoneum	Joint aspirate	Epithelioid granuloma with giant cells without necrosis	^1–2^[Am-Eto-Lzd-Mfx-Pza]/ ^3^[Am-Emb-Eto-Lzd-Mfx-Pza]/ ^4–5^[Am ^†^-Bdq-Emb-Eto ^†^-Lzd-Mfx-Pza]/ ^6–23^[Bdq-Cfx-Emb ^†^-Lzd-Mfx-Pza]	Joint washing	Success ^§^ (1 year)
21	M, 35	HIV	Ongoing pulmonary MDRTB treatment for two years	2018	L2–S1, psoas abscess	-	Psoas abscess	ND	^1–5^[Bdq-Cfz-Lzd-PAS]/ ^6–7^[Bdq-Cfz-Dlm-Lzd-Mpm/Clv]/ ^8–14^[Bdq-Cfz-Cm-Dlm-Lzd-Mpm/Clv] ^15–?^[Bdq-Cm-Dlm-Lzd]-(duration not known due to loss of follow-up)	Laminectomy	Not evaluated (loss of follow-up after 15 months of treatment)
22	M, 8	None	None	2018	Right hip joint, gluteus abscess	Lymph nodes	Peri-joint abscess	ND	^1–18^[Cs-Dlm-Eto-Lfx-Lzd]	Peri-joint and gluteus abscess drainage	Success (no post-treatment follow-up)
23	M, 39	HIV	None	2018	Left seventh rib, both iliac bones	Lung, lymph nodes, pleura, liver, spleen	Pulmonary sample	ND	^1^[Am-Emb-Lzd-Mfx-Pza]/ ^2–3^[Am-Dlm-Emb-Lzd ^†^-Mfx-Pza]/ ^4–19^[Dlm-Mfx-Pza]	No surgery	Success (6 months)

F, female; M, male; TB, TB; Am, amikacin; Bdq, bedaquiline; Cfz, clofazimine; Cm, capreomycin; Cs, cycloserine; Dlm, delamanid; Emb, ethambutol; Eto, ethionamide; Inh, isoniazid; Km, kanamycin; Lfx, levofloxacin; Lzd, linezolid; Mpm/Clv, meropenem/clavulanate; Mfx, moxifloxacin; Ofx, ofloxacin; PAS, para-aminosalicyclic acid; Pto, protionamide; Rif, rifampin; Pza, pyrazinamide. * Treatments are shown as number of months followed by drugs administered in each phase. Different phases are divided by “/”. ^†^ Discontinued due to toxicity. Myelosuppression led to the withdrawal of linezolid (*n* = 2) and PAS (*n* = 1). Peripheral neuropathy was observed with linezolid in 2 cases. Gastro-intestinal symptoms were encountered with PAS (*n* = 2), ethionamide (*n* = 1) and pyrazinamide (*n* = 1). Ethambutol was withdrawn after retrobulbar optic neuritis in 2 cases, and both ethambutol and linezolid were withdrawn for the same reason in 2 cases. Two cases of renal insufficiency and one of ototoxicity occurred with amikacin. Bedaquiline was discontinued because of QT prolongation (*n* = 1). Hepatic disorders occurred with pyrazinamide, rifampin and moxifloxacin (*n* = 1 in each case). ^‡^ Toxicity without discontinuation. ^§^ Patients who would have been classified as treatment failure if we had applied the fourth criterion (discontinuation of ≥2 drugs) during the consolidation phase (Appendix A).

**Table 3 microorganisms-10-01215-t003:** Resistance to anti-TB drugs (%) among strains isolated from patients with multidrug-resistant osteoarticular TB in France (*n* = 23) and among all strains isolated from MDR-TB patients (*n* = 863), from 2007 to 2018.

Drug	MDR-OATB ^1^ *n* (%)	MDR-TB *n* (%)	*p*-Value
**Rifampin**	18 (78) high level/5 (22) low level ^2^	837 (97) high level/26 (3) low level	<0.01
**Protionamide**	15 (65)	673 (78)	0.06
**Streptomycin**	15 (65)	811 (94)	<0.01
**Ethambutol**	12 (52)	681 (79)	<0.01
**Pyrazinamide**	6 (26)	379 (44)	0.01
**Kanamycin**	3 (13)	267 (31)	<0.01
**Cycloserine**	2 (9)	215 (25)	<0.01
**Amikacin**	2 (9)	121 (14)	0.38
**Ofloxacin**	2 (9)	259 (30)	<0.01
**Moxifloxacin**	2 (9)	259 (30)	<0.01
**Bedaquiline ^3^**	1 (4)	34 (4)	1
**Capreomycin**	1 (4)	138 (16)	<0.01
**PAS**	1 (4)	155 (18)	<0.01
**Linezolid**	0 (0)	0 (0)	1

^1^ Two strains were XDR according to the definition before 2021 [8]. ^2^ Mutations in *rpoB* were: L452P, D435Y, L430P, H445L and M433I [13]. ^3^ Bedaquiline DST was not implemented before 2014, therefore the available DST results did not include all the MDR-RT strains isolated during the study period.

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
