# Peer review of "Clinical Features and Outcome of Multidrug-Resistant Osteoarticular Tuberculosis: A 12-Year Case Series from France"

_microorganisms, 2022, doi:10.3390/microorganisms10061215_

Round 1

Reviewer 1 Report

The osteoarticular MDR-TB is rare and very difficult to diagnose. This case series is very interesting and relevant for publication. However, the authors might consider the following comments to improve the manuscript

1-what is missing in this introduction is the epidemiological data on osteoarticular TB in France. If there is not, it is an additional reason justifying this article (it must therefore be added)

2-Line 97.98; 103-104 the authors included cases between 2007 and 2018. Therefore I am not sure that the new definition of Pre XDR-TB(released in 2021) is applicable. Please revise

3-Line 128-130: a flowchart would be fine

4-Table 1: No need to add the diabetes comorbidities row (since there is no case)

5-The title of table 3 should specify the number of patients

6-Furthermore, the absolute number is missed in the table, no one can know how the percentages were obtained. Please improve the presentation of the table by using the common way to present the table in an epidemiology article.

Reviewer 2 Report

This is a very good paper reporting a series of patients with osteoarticular TB (OATB) due to multidrug-resistant strains (MDR-TB) and the outcome of treatment. The paper is clearly written and contains many useful informations but could be improved by clarification of several points:

1. Introduction, line 76: please indicate which proportion of TB are extrapulmonary (suggestion "about one quarter")

2. Intro. line 89: is there any evidence that the proportion of MDR-TB strains causing OATB is different from the proportion causing pulmonary TB? In other words, are extrapulmonary tissues more sensitive to MDR-TB strains than the lungs?

3. Materials, line 103: although most readers will know what is XDR-TB, please define (you have defined MDR-TB, why not XDR-TB?)

4. Tab 1. a) contrary to the common knowledge, half of your patients with OATB were from Africa and not, as expected, from Eastern Europe. A comment in the discussion would be welcome (effet of large population from African origin?)

5. Tab 1, b) I suggest to add at the end of the table the number and % of OATB with lung involvement (9, if I count correctly), as this may be a very important diagnostic clue (easier collection of samples)

6. Line 176: seventeen

7. Results, line 194: unclear: were the patients who needed the replacement of more than 2 drugs cured or not?

8. Discussion, line 230: fully agree with the comment that the radiological appeasrance should not be the criterion for treatment discontinuation (likewise in pulmonary TB!), but did you use other criteria apart from the treatment duration?

9. Discussion, line 237: the drug resistance could have been acquired against the drugs used in both situations, against first-line and second-line drugs

10. Discussion, line 252. Although we are perfectly aware that the regimens were tailored to the drug susceptibily, it would be interesting to know which proportion of the patients were treaeted with the main second-line drugs (quinolones, linezolid, injectables - in particular  carbapenems) and from when you started using bedaquiline. Did you consider high-dose isoniazid in some patients, as you seem to have used high-.dose rifampicin? 
